## [Decision Letter · Decision Letter 0]

17 Nov 2025

Dear Dr. Watanabe,

Thank you for submitting your manuscript to PLOS ONE. After careful consideration, we feel that it has merit but does not fully meet PLOS ONE’s publication criteria as it currently stands. Therefore, we invite you to submit a revised version of the manuscript that addresses the points raised during the review process.

We look forward to receiving your revised manuscript.

Kind regards,

Mohanad Alkhodari

Academic Editor

PLOS ONE

Journal Requirements:

“This work was supported by a JSPS Grant-in-Aid for Research Activity Start-Up [grant number: JP22K21220] and a Kinjo Gakuin University Research Grant.”

6. In the online submission form, you indicated that “Data cannot be shared publicly, because many authors contributed to obtain data. Data are available from Okayama Univ. for researchers who meet the criteria for access to confidential data. Contact to the corresponding author, Shogo Watanabe (watanabe1224@okayama-u.ac.jp).”

**Additional Editor Comments:**

The paper requires major revision as requested by Reviewer 2.

Reviewers' comments:

Reviewer's Responses to Questions

**Comments to the Author**

1. Is the manuscript technically sound, and do the data support the conclusions?

Reviewer #1: Yes

Reviewer #2: No

2. Has the statistical analysis been performed appropriately and rigorously?

Reviewer #1: Yes

Reviewer #2: Yes

3. Have the authors made all data underlying the findings in their manuscript fully available?

Reviewer #1: Yes

Reviewer #2: No

4. Is the manuscript presented in an intelligible fashion and written in standard English?

Reviewer #1: Yes

Reviewer #2: No

Reviewer #1: Very informative and sound manuscript. The research include the required statistics as well as the limitation of the study. The conclusion is written based upon the analysed data and results. The recommendation to extend the number of the studied animals should be included

Reviewer #2: The authors investigate whether soluble guanylate cyclase (sGC) stimulators and RAS inhibitors protect against HFpEF. This study aimed to evaluate the preventive effects of RAS inhibitors captopril (Cap) and/or sacubitril/valsartan (Sac/Val) and sGC stimulator vericiguat (Ver) on HFpEF progression. HFpEF was induced in 8-week-old male Wistar rats through intake of L-arginine methyl ester and a high-fat diet. Results showed that the survival rate after 8 weeks of treatment was 100% in the normal diet (Cont group), Cap, and Sac/Val groups, whereas it was approximately 20% in the HFpEF and Ver groups. No significant differences in the left ventricular systolic function were found. In addition, histochemistry revealed that myocardial hypertrophy and interstitial fibrosis obviously increased in the HFpEF group but not in the Cap and Sac/Val groups compared with the Cont group. Furthermore, RNA sequencing analysis showed that the expression of genes related to inflammatory response, hypertrophy, and extracellular matrix–receptor interaction increased in the HFpEF group and decreased in the Cap and Sac/Val groups.

The manuscript is interesting. However no functional effect of early administration of Cap or Sac/Val has been showed. My suggestion is to performed among groups:

Invasive Hemodynamics (Pressure–Volume Loop Analysis) or in alternative Treadmill endurance test

**Do you want your identity to be public for this peer review?** For information about this choice, including consent withdrawal, please see our Privacy Policy

Reviewer #1: **Yes:** Hala Mounir Agha

Reviewer #2: No

---

## [Author Response · Author response to Decision Letter 1]

23 Nov 2025

We revised the manuscript and author information according to the reviewers' and editors' instructions.

---

## [Decision Letter · Decision Letter 1]

9 Dec 2025

Early administration of renin–angiotensin system inhibitors improves survival and cardiac remodeling in heart failure with preserved ejection fraction

PONE-D-25-48814R1

Dear Dr. Watanabe,

We’re pleased to inform you that your manuscript has been judged scientifically suitable for publication and will be formally accepted for publication once it meets all outstanding technical requirements.

Kind regards,

Mohanad Alkhodari

Academic Editor

PLOS One

Additional Editor Comments (optional):

Based on the recommendations of two reviewers, the paper is to be accepted in its current form.

Reviewers' comments:

Reviewer's Responses to Questions

**Comments to the Author**

Reviewer #2: All comments have been addressed

2. Is the manuscript technically sound, and do the data support the conclusions?

Reviewer #2: Yes

3. Has the statistical analysis been performed appropriately and rigorously?

Reviewer #2: Yes

4. Have the authors made all data underlying the findings in their manuscript fully available?

Reviewer #2: Yes

5. Is the manuscript presented in an intelligible fashion and written in standard English?

Reviewer #2: Yes

Reviewer #2: The manuscript is rteally improved and all the questions have been aswwered. No further modifications are requested

**Do you want your identity to be public for this peer review?** For information about this choice, including consent withdrawal, please see our Privacy Policy

Reviewer #2: No

---

## [Editor Report · Acceptance letter]

PONE-D-25-48814R1

PLOS One

Dear Dr. Watanabe,

I'm pleased to inform you that your manuscript has been deemed suitable for publication in PLOS One. Congratulations! Your manuscript is now being handed over to our production team.

Kind regards,

on behalf of

Dr. Mohanad Alkhodari

Academic Editor

PLOS One